# A VM/Containerized Approach for Scaling TinyML Applications

## ABSTRACT

Although deep neural networks are typically computationally expensive to use, technological advances in both the design of hardware platforms and of neural network architectures, have made it possible to use powerful models on edge devices. To enable widespread adoption of edge based machine learning, we introduce a set of open-source tools that make it easy to deploy, update and monitor machine learning models on a wide variety of edge devices. Our tools bring the concept of containerization to the TinyML world. We propose to package ML and application logic as containers called Runes to deploy onto edge devices. The containerization allows us to target a fragmented Internet-of-Things (IoT) ecosystem by providing a common platform for Runes to run across devices.

## CCS CONCEPTS

• **Computing methodologies → Machine learning**; • **Computer systems organization → Embedded and cyber-physical systems**.

## KEYWORDS

Edge AI, TinyML, Neural Networks, Rune, Containerization, Deep Learning, Internet of Things (IoT)

## 1 INTRODUCTION

In the past decade, deep neural networks (DNNs) have become the state-of-the-art technique for many applications in domains like computer vision, audio processing, robotics and natural language processing [20]. A disadvantage of these models is that they are typically computationally expensive to use. A Resnet50 network for example, a common model for image classification, requires around 100Mb to store its parameters and 4 GFLOPs to process a single 224x224 RGB input image [17]. The most common approach is to use high end Graphical Processing Units (GPUs) to train and evaluate the models in a cloud-based system. The alternative of deploying neural network models on edge devices is often more attractive since this can result in a lower latency and increased robustness as no network connection to the cloud is required. In addition this is also more privacy friendly compared to cloud based deployments as no privacy sensitive data ever leaves the local device.

Deep neural networks benefit from large amounts of rich input data. This makes them especially useful to process the giant volume of data which has been generated by billions of Internet of Things (IoT) devices. The number of these devices is estimated to grow to 22 billion by 2025 [9]. Uploading all this data to the cloud for analysis would require significant bandwidth resources. Instead, a more decentralized approach based on edge deployments is much more scalable. A major challenge is however posed by the diversity of hardware platforms used in an IoT ecosystem. All these devices

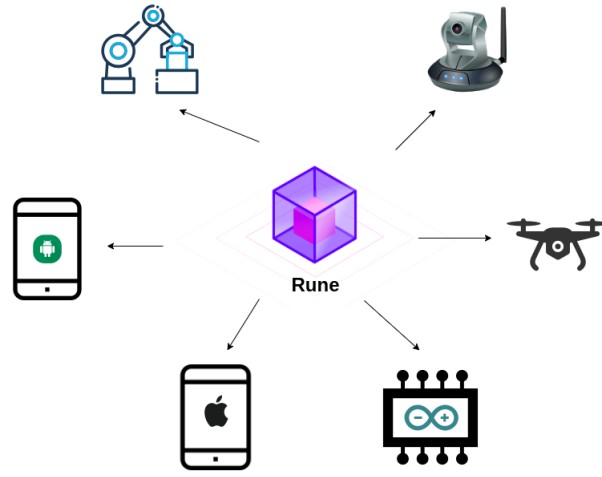

**Figure 1: Runes package ML models and additional code into a container that can be deployed on various hardware platforms using Hammer.**

have different sensors, different computational resources and different costs associated with network connectivity. It is therefore not easy to dynamically deploy models to devices. In this paper, we introduce a software framework that will make it possible to easily package, distribute and monitor TinyML applications on heterogeneous IoT platforms.

Containerization of applications in the cloud has provided reliability, scalability and security as it allows us to dynamically place software components on different hardware devices. We aim to bring these concepts to the TinyML world. We propose to package ML and application logic as containers called Runes to deploy onto edge devices. The containerization allows us to target a fragmented Internet-of-Things (IoT) ecosystem by providing a common platform for Runes to run on different devices. We provide a declarative configuration for developers to build model pipelines that can be deployed across multiple classes of devices. Security is provided by creating abstracted interfaces to hardware capabilities which facilitates simulation, testing and continuous integration.

Currently, edge applications need to be compiled and built directly against the device software/hardware. Containerization decouples the Rune bytecode from the hardware, device capabilities, sensors, and processing architectures. With Rune, the model is built once for a device and can then run on compatible devices, reducing development time and making the application more portable while reducing the overall development time across devices. Rune also supports provisioning which guarantees that only the required bytecode operations and sensor access is made available. Runes have low-level APIs that can be programmed against a variety of languages (C/C++, Go, Rust, etc), making it easy for developers to

bring their machine learning models into production. Similar to the Docker registry, Runes can be made available in a community registry. These Runes can then be reused or extended by adding additional processing components.

In addition to the Rune containerization, we also release a tool call Hammer. Hammer is used to deploy Runes over the air via Bluetooth Low Energy (BLE), a serial connection, WiFi or other connectivity protocols. Hammer can be used to ship complex applications confidently by eliminating developer computers as the source of origin. Targeting multiple devices will become simple using configurations instead of code.

The remainder of this paper is organized as follows. In section 2 we give an overview of the related work in optimizing neural networks for edge deployments and the software tools that are available to do so. We also give a very brief overview of containerization and Docker as these formed the inspiration for Rune. In section 3 we introduce our different tools and give more details about the design choices. We benchmark the overhead of the containerization layer in section 4. Finally, we conclude and give some pointers for further research and development in section 5.

## 2 RELATED WORK

In this section we give an overview of some interesting technologies that can make neural networks more efficient to use on resource constrained edge devices. We also discuss existing software platforms that target edge devices and provide some background on containerization and virtualization.

### 2.1 Efficient neural network architectures

There is a lot of active research into making deep neural networks more efficient to use on smartphones and other edge devices. A first family of approaches reduces the model size by pruning weights of the network. Various strategies have been proposed that select the weights to remove based on the sensitivity of the final objective function to that weight [21] or on the magnitude of the weight [16]. It is also possible to include additional constraints in the pruning algorithm such as the energy consumption of the different layers [25].

It is well known that full precision floating point numbers are not needed for neural network weights and activations. Instead 16 bit [11] or even 8 bit numbers [23] are usually sufficient. By quantizing the weights and activations, we can replace the floating point arithmetic with integer operations that are more efficient in hardware [15]. Other works further reduce the precision of the weights to 4 bits [10] or even all the way down to one bit [19]. In this case, most of the operations are replaced with logical operations that are very efficient in hardware [19].

Other approaches replace the expensive convolutional layers with depthwise separable convolutions [18] that split a convolutional kernel into two separate kernels that do two convolutions: the depthwise convolution and the pointwise convolution. This reduces the amount of computations and still results in a high accuracy. Other works further reduce the computational cost by replacing the

first convolution with a cheap channel shuffle operation. Instead of manually designing efficient architectures, it is also possible to perform an automated architecture search for architectures that result in a good computational cost-accuracy trade-off [26].

### 2.2 Software frameworks for neural network deployment on edge devices

Various software platforms are being actively developed to enable neural network execution at the edge. Tensorflow Lite is a set of tools to help developers run TensorFlow models on mobile, embedded, and IoT devices. It enables on-device machine learning inference with low latency and a small binary size [13]. To target even more constrained devices such as microcontrollers. Tensorflow lite micro is designed to run machine learning models on devices with only few kilobytes of memory. The core runtime just fits in 16 KB on an Arm Cortex M3 and can run many basic models. Similar to Tensorflow Lite, Apple's CoreML framework [3] allows developers to embed a neural network in a mobile app. In addition to running inference, Core ML also makes it possible to train or fine-tune models on the user's device. Core ML optimizes on-device performance by leveraging the CPU, GPU, and Neural Engine while minimizing its memory footprint and power consumption. The Neural Engine is a specialised chip in Apple devices that can evaluate neural networks in a more energy-efficient manner than using the main CPU or the GPU. Recently, a beta version of Pytorch mobile has been released [7]. Pytorch mobile allows developers to take a trained pytorch model, quantize it to 8 bit integer representations and save it in an optimized format to use in an Android or IOS app. An especially interesting software framework in this space is Apache TVM [12]. TVM can take models trained in different frameworks such as Tensorflow or Pytorch, compile them into minimum deployable modules and run them efficiently on different hardware backends such as CPUs, GPUs, microcontrollers or even FPGAs.

There are a few software tools available that can aid in deploying neural networks in a production environment. TensorFlow Serving [8] is a flexible, high-performance serving system for machine learning models. It takes a trained Tensorflow model, deploys it on a server and provides a REST api to access the functionality of the model. Similarily, MLFlow [5] can deploy models trained in different frameworks to Apache Spark, Azure ML and AWS SageMaker. Another interesting option is the NVIDIA Triton Inference Server [6], an open source inference serving platform that can deploy trained AI models from any framework on any GPU- or CPU-based infrastructure.

There are also a few solutions that can be used to automate edge deployments. DIANNE [14] is a modular framework that can dynamically distribute (parts of) deep learning models over multiple edge and cloud devices. Another option is the IBM Edge Application Manager (IEAM) [4] that provides a Model Management System (MMS) that can asynchronously update machine learning models running on Edge nodes. Currently however, it does not support deploying models on devices such as microcontrollers. Similarly, Microsoft provides the Azure IoT Edge [2], a fully managed service built on Azure IoT Hub. It can deploy ML workloads on Internet of

Things (IoT) edge devices. It also uses a container-based approach. Amazon also has a solution, called AWS IoT Greengrass [1]. This makes it easy to perform machine learning inference locally on devices, using models that are created, trained, and optimized in the cloud. It can target common devices such as Intel Atom, NVIDIA Jetson, and Raspberry Pi but like most of the previous platforms, it is not suitable for extremely constrained devices such as microcontrollers. The tools that we present in this paper can be used for the same use cases but we also target devices with very few resources such as Arduino microcontrollers.

## 2.3 Containerization and virtualization

In the last few years, containerization and microservices-based architecture design have become one of the most popular approaches for cloud based application development. Containerization involves encapsulating software code and its dependencies in a container so that it can easily be deployed on any infrastructure. Containers are a special approach to virtualization. Containers allow you to run an application isolated from its host operating system, without having to spin up an entire virtual machine (VM) for each application. By breaking up software into minimal components called "microservices" that can each be deployed in a container, a large software system can be split up into multiple components which makes it easier scalable, testable and reduces the complexity of development and integration. Compared to a VM, a container is "lightweight" as it shares the machine's operating system kernel and does not require the overhead of launching a full operating system for each application. Containerization is especially attractive as it allows applications to be "written once and run anywhere."

Docker [22] is the most well known example of a containerization framework. To create a new Docker container, the developer first writes a Dockerfile that lists all the steps needed to build an image. Dockerfiles typically refer to a parent Dockerfiles. A Docker image inheriting from another Docker image is capable of all functionality of the parent and can add its own components. Docker images can be shared on Docker Hub allowing other developers to reuse or extend existing images. Runes follow the same approach as Docker images. They allow a developer to package ML models and the additional code. The resulting images can then be deployed to different devices or can be shared on a model hub, enabling other developers to reuse or extend them.

## 3 DEPLOYING MODELS ON EDGE DEVICES

As machine learning techniques mature and their usefulness for different fields becomes apparent, there is an increasing need for tooling that supports developers to use the models in production. This is sometimes named "MLOps" after "DevOps". "MLOps" techniques aim to automate the process of (re)training a model, validating it and deploying the model for inference. Currently, the models are typically deployed to a cloud infrastructure and access is provided through a REST api. As mentioned in the introduction, deploying models on edge devices is often an attractive solution

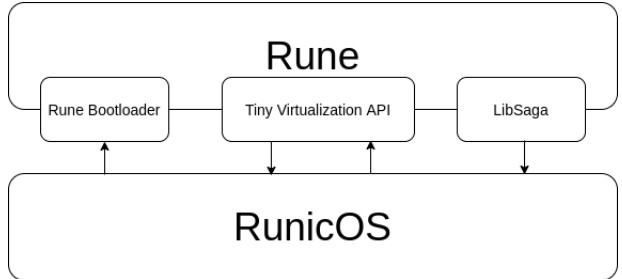

**Figure 2: The edge device runs RunicOs that boots Rune containers and provides APIs to access sensors and memory. LibSaga is used to monitor the state of the Rune.**

since it reduces the latency, does not require a high bandwidth network connection and is privacy friendlier. In addition to the limited computational resources on these devices, a major bottleneck for widespread adoption is the IoT platform fragmentation. The lack of interoperability and of common technical standards makes it difficult to develop applications that work consistently between different inconsistent technology ecosystems. To solve this, we introduce Rune and Hammer. The next sections explain both tools in more detail. Both tools are developed in the Rust programming language and can be found at our Github page: https://github.com/hotg-ai.

## 3.1 Rune

Just like Docker images package software together with its dependencies, we propose to package ML and application logic into a container called Rune. Runes can then easily be deployed onto different edge devices. By providing a common platform for Runes to run, we can target the fragmented IoT platforms, making it easier for developers to deploy their models on real world devices.

Figure 2 shows the interaction between the different components needed to run a Rune. The device runs RunicOs. A minimal operating system that can boot a Rune and exposes hardware capabilities with a common access layer. Rune containers are booted using the Rune container bootloader which is initiated by the RunicOS. RunicOS is a collection of APIs that provide access to sensors, communication stacks, and memory or filesystems. The tiny virtualization API enables encapsulation and execution of user space bytecode in Rune. Finally, LibSaga is the library that provides communication and monitoring of Rune health. In future versions, LibSaga will also allow debugging and benchmarking of the models, making it possible for the developer to monitor the throughput and system load of the Rune.

The operating systems is also responsible for guaranteeing that only the required bytecode operations and sensor access is made available to the Rune. As IoT devices equipped with cameras, microphones and other sensors are deployed in potentially privacy sensitive environments such as people's homes or offices, we have to make sure that a Rune developed by a third party only has access to the resources needed for a certain task. The RunicOS enforces these limitations using a permissions system, following the Principle of least privilege, which states that the running software should

**Listing 1: Example Rune file**

```
1  FROM runicos/base
2  CAPABILITY AUDIO audio --hz 16000 --samples 150 --sample-size 1500
3  PROC_BLOCK runicos/fft fft
4  MODEL ./example.tflite model --input [150,1] --output 1
5  RUN audio fft model
6  OUT serial
```

only have access to the minimal amount of resources needed to do its job. A model deployed for speech recognition for example should have no access to a camera attached to the device.

As the IoT ecosystem is characterized by a huge variety in available devices, there is no guarantee that a certain Rune can run on all possible devices. Different devices might have different computational resources or different sensors. The RunicOS is also responsible for checking these requirements prior to launching the Rune. If a sensor is available and the Rune has permission to use it, the RunicOs provides an abstract API to access the sensor. The same Rune can then easily be deployed on different devices, with different variants of a similar sensor.

Similar to a Dockerfile, a Rune is configured with a Runefile, an example is shown in listing 1. The Runefile always starts with a FROM instruction which defines the base layer on which the container is built. When processed, the FROM instruction verifies that RunicOS is present and ready for loading of a Rune and that its dependencies are satisfied. Runes are composable, developers may extend prebuilt containers by referring to these containers with the "FROM" instruction.

On line 2, we configure what sensors are required for this Rune to work. In this case, we require a microphone input which is labelled as "audio" for reference further in the Runefile. We can optionally pass additional arguments such as the sampleRate. Line 3 defines a processing block "fft". Processing blocks are arbitrary C++, Rust or webassembly modules that can be used to preprocess sensor outputs for models. In this case, we use a default block "runicos/fft" which calculates the discrete Fourier transform of the input sequence, converting the audio signal to the frequency domain. The neural network model is defined on line 4. In this case it is provided as a Tensorflow lite model. In future versions of Rune, support will be added for other frameworks such as CoreML or Pytorch. Line 5 then combines all blocks, it defines where the input data should come from and what processing blocks need to be executed before the model is evaluated. Finally, the output of the model is writen to a serial connection (line 6).

Once the developer has defined the RuneFile, the Rune Container can be built

```
$ rune build <Runefile>
```

Behind the scenes, the "rune" command generates a Rust project based on the RuneFile. The RuneFile is translated into Rust source code containing two important functions: "_manifest()" and "_call()".

These functions are used to initialize a Rune. The first step (manifest step or manifest stage) is to verify the requirements of the Rune. The "_manifest()" function returns a manifest struct as shown in listing 4. This is used by the RunicOS to ensure that the hardware has the right capabilities and memory space to run the model described. Once the Rune has been successfully initialized, the "_call()" API can be used to run the Rune. The reason to have a two-step process is to ensure that the device will not fail when the "_call()" API is executed, as the process has already determined that dependencies required to run the TinyContainer are present, by processing the manifest.

**Listing 4: The Manifest struct describes the requirements of the model**

```
1  struct Manifest{
2      capabilities; Vec<CapabilityRequest>,
3      out: OUTPUT,
4      models: vec<Model>
5  }
```

This generated Rust project includes the source code for all the building blocks of the container such as the code to access the sensors, a compiled version of the model and all the code for the processing blocks. This project is then compiled into a WebAssembly module that can be launched on the edge device. WebAssembly (Wasm) is a binary instruction format for a stack-based virtual machine. It is an open standard that defines a portable binary-code format for executable programs. Originally, the purpose of WebAssembly was to enable high-performance applications on web pages. Web applications can use Wasm alongside HTML, CSS, and JavaScript native in a browser. Developers can compile C++ (or any other LLVM-supported language such as D or Rust) source code into a binary file which runs in the same sandbox as regular JavaScript code. Since WebAssembly's runtime environments are low level virtual stack machines, they are not limited to web applications and can also be embedded into host applications or can be launched in standalone runtime environments. A very interesting application is to use Webassembly to deploy applications on IoT and edge devices [24] where the sandboxing functionality of Webassembly can offer high security while still having a very high performance.

Once the Webassembly module is generated, the Rune can be deployed on an edge device, either manually or using Hammer as described in the next section.

**Listing 2: Hammer can list the devices that are available as targets for deployment**

```
$ hmr targets ls
Target                           Type        Name            Available
------------------------         ---------   ------------    ---------------
/dev/tty.usbmodem14301           WIFI        Portenta H7     True
2b99c594-c904-4133               WIFI        Portenta H7     True
```

**Listing 3: Hammer can deploy a Rune to a target device**

```
$ hmr targets cast -t 2b99c594-c904-4133 ./microspeech.rune

Deploying ./microspeech.rune to target 2b99c594-c904-4133
Verifying provider: 100%
Provider with fqdn=arduino:mbed found
Uploading rune: 100%
Verifying rune: 100%
Capabilty: WIFI OK: 100%
```

## 3.2 Hammer

Hammer is the tool that will help deploy Rune image on one or more devices, either wirelessly or over the wire. It helps operate, deploy and test Rune based apps on devices as well as monitor the health of Rune based apps. Hammer abstracts away the different communication protocols (BLE, WiFi, Serial, ...) needed to communicated with the devices. Listing 2 shows devices that are available for deploying Runes as a result of the "hmr target ls" command. For each device, a location of the device is listed under the "Target" heading. The type of communication protocol used to communicate with the device is listed under the "Type" heading. A name of the device as provided by the manufacturer is listed under the "Name" header and availability of the device is listed under the "Availabilty" header.

Once the target device has been identified, a Rune can be deployed to that device with the "hmr targets cast" command as shown in listing 3.

## 4 EXPERIMENTS

As explained in the previous sections, Rune allows us to create "write once, run anywhere" containers that package machine learning models and the additional required code in a WebAssembly package. This is then launched on the device as a container where the RunicOS provides APIs to access sensors or other functionality. The RunicOS abstracts away the device specific functionality, making it possible to run the same Rune on a different device. The price we pay for this flexibility is the overhead involved in the containerization. In this section, we benchmark this overhead on an Arduino Portenta H7 device.

We run the same code, once directly compiled for the device, once as a Rune with Rust serialization, once as a Rune with Proto-Buf serialization, and measure the total time needed for running a simple model that estimates the sine function. We calculate the overhead for 1000 to 1 million calls to the model.

The results are summarized in Figure 3. The Figure shows the

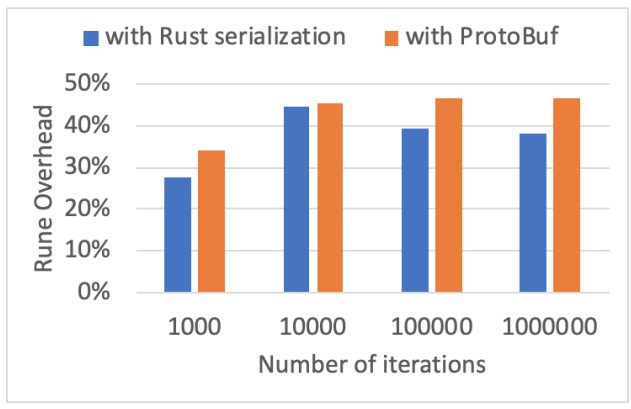

**Figure 3: Benchmarking results. The same code implementing a sine model was run for 1000 to 1 million iterations, once directly compiled for the device, once as a Rune with Rust serialization and once as a Rune with ProtoBuf format. The % overhead in running time (equation 1) was recorded for each experiment.**

overhead in CPU time, in percentage terms, due to using a Rune container to run the model:

$$overhead = (t_{Rune} - t_{native})/t_{native} \tag{1}$$

The measured overhead varies from 28% to 45%, and is around 40% at 100k to 1 million iterations. The variation at the lower number of iterations can be explained by statistical variation in the computation times.

Using protobuf in the experiments introduced a slight increase in computation time. We expect our data sharing layer to be the most likely culprit to the overhead, and expect future improvements from optimizing our protocol with ProtoBuf format.

# 5 CONCLUSION AND FUTURE WORK

In this paper, we introduced Rune and Hammer, two tools that will make it easier for developers to deploy their machine learning models on a variety of hardware devices. Even though both tools can be used to deploy the models on cloud infrastructure, their main goal is to enable deployment on edge devices with limited computational resources. With Rune, we defined a containerization based approach that packages ML models and the required additional code into a small WebAssembly package that can be deployed to different devices. The containerization allows us to run the same Rune on different devices, without requiring any additional configuration. Runes are configured by Runefiles, similar to Dockerfiles. This makes it easy to reuse or extend existing Runes for new tasks. We also introduced Hammer which can push Runes to a variety of devices, over a wired or wireless connection.

We believe the Rune and Hammer tools can be crucial building blocks to enable widespread adoption of on-device machine learning. In future work, we will keep improving both tools by adding support for more devices, sensors and machine learning frameworks. We will also build a Kubernetes-like system that uses Rune and Hammer to automatically deploy, scale and manage large machine learning deployments on heterogeneous IoT hardware. This will allow multiple devices to be in a redundant state for service continuation. The workload resilience can come from multiple cores of a single device (vertical scaling) as well as from multiple devices in a meshed setting (horizontal scaling).

All tools will be released as open source software on our github page[1]. We aim to build a strong community supported ecosystem and are actively looking for contributors.

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
