# OpenReview forum: "A VM/Containerized Approach for Scaling TinyML Applications"
_tinyml.org/tinyML/2021/Research_Symposium — tinyML 2021 Poster_

### Official Review · AnonReviewer3 · 2021-01-07

**Overall Merit Score:** 2

**Brief Summary:**

The paper describes how entire processing pipelines including ML model inference can be packaged into containers called "Runes".
These containers resolve external code dependencies and may target multiple different processing platforms.
Target platforms have to provide a runtime system consisting of a standardized operating system called "RunicOS" and an interpretation engine for WebAssembly code.
The overhead in processing time over a native compile to the device hardware is measured and discussed.
The processing platforms targetted by this approach are rather of Arduino type than Raspberry PI or alike.
Moreover a tool called "Hammer" is mentioned that facilitates deploying such containers over standard wired and wireless interfaces.

**Detailed Comments:**

This paper would potentially be more interesting for a conference on general embedded programming techniques.
It does not give an answer on the obious question of memory overhead, which is particularly critical for the targetted systems.

**Paper Strengths:**

* The paper explains adequately the state of the art for generating efficient neural network architectures for edge devices.
* Moreover it summerizes well the relevant software frameworks for deploying such models on edge devices.
* It addresses a the very relevant problem that the great diversity of embedded processing platforms requires deep expert knowledge for mapping an application to a device.
*Figures for processing overhead are presented and discussed.

**Paper Weaknesses:**

* The paper deals with software techniques that are not very ML specific (but can be applied to ML too). Actually for the ML part frameworks like TFLite Micro do already provide a platform independent serialization with Flatbuffers, which is similar to ProtocolBuffers but more efficient for the task.
* In capter 4 the overhead in execution time is discussed, but the overhead in memory footprint would be as desirable to know. I would expect it to be quite high.
* The benchmark case "a simple model that estimates the sine function" should be referenced. A more realistic choice of model would be recommended (e.g. "wakeword").
* The deployment tool "Hammer" is only briefly mentioned which is rather distracting than adding value.

**Poster (If Paper Is Rejected):**

1: Yes, ok for poster sesion to nurture work

**Reviewer Confidence:**

3: The reviewer is fairly confident that the evaluation is correct

---

### Official Review · AnonReviewer2 · 2021-01-26

**Overall Merit Score:** 1

**Brief Summary:**

Putting ML model and runtime in Linux container, called Rune

**Detailed Comments:**

See above

**Paper Strengths:**

BLE orchestration is one of features needed in TinyML, but this part should be an independent paper of the other part of Linux Container.

**Paper Weaknesses:**

It's a little big difficult to call this Linux Container based orchstraion for TinyML since most of ML service orchestration on Cloud is usually done by Linux Container. In some cases, still there are even some hardware heterogenity included. For me, what's special for TinyML from software architecture perspective is that it cannot fit in places Linux fits in. Once Linux doesn't fit all the pain starts, where you cannot re-use the  established Cloud ML solutions at all. This paper might be useful for other conference but not for this one.

**Poster (If Paper Is Rejected):**

1: No, paper is below bar for poster as well

**Reviewer Confidence:**

5: The reviewer is absolutely certain that the evaluation is correct and very familiar with the relevant literature

---

### Official Review · AnonReviewer4 · 2021-01-29

**Overall Merit Score:** 2

**Brief Summary:**

The paper describes a container-like technology, similar to Docker, suitable for micro-controller based devices.   The system includes an operating system, Rune, and a loading/deployment tool called Hammer.  The authors show some experimental results on the run-time overhead relative to a native baseline.

**Detailed Comments:**


* How does the proposed approach compare to commercial efforts like Edge Impulse or Arm's MBED?
* Section 2.3 - The authors state that "Containerization ... allows applications to be 'written once and run anywhere.'"  But for typical container approaches (e.g. Docker), that relies on all hardware targets being binary-compatible and running the same OS.
* It is not clear whether the Rune application is intended to be a relatively enclosed function like a single ML model, or a more encompassing application, like an entire watch interface.  Some examples spanning the range would help clarify.
* Can the Rune system deploy a project to different MCU and DSP architectures?  If so, it would be good to demonstrate that.
* Embedded software often works tightly with the hardware in a way that makes it difficult to abstract away the hardware.  How are variations in hardware (different peripherals, ADC interfaces, etc.) handled?  Again, a demonstration would help here.
* pg. 4 - Is there a reference for web assembly (Wasm) you can cite?
* The biggest issue with this work is that it is not clear what advantage Rune/Hammer offers with respect to currently available approaches.  For example, if the main objective is to run ML Models on varied devices, how is Rune superior to just having a TFLite-Micro interpreter available on all the target devices?  This approach forces an OS choice on the system designer, so the advantages should be substantial.


**Paper Strengths:**

* The paper is well-organized and easy to follow.
*The authors address a significant problem in the difficulty of porting small embedded applications to different hardware.
* The benchmarking results showing the overhead of the proposed solutions are good information.

**Paper Weaknesses:**

* The biggest weakness is that it is not clear what overall advantage the proposed technology brings.
* There are some details about how flexible the proposed solution is that are not clear.

**Poster (If Paper Is Rejected):**

1: Yes, ok for poster sesion to nurture work

**Reviewer Confidence:**

4: The reviewer is confident but not absolutely certain that the evaluation is correct

---

### Official Review · AnonReviewer1 · 2021-01-30

**Overall Merit Score:** 2

**Brief Summary:**

This paper describes a system that virtualizes running machine learning models on embedded devices, by providing an environment with common functionality across a wide variety of different platforms.

**Detailed Comments:**

Detailed comments are inline above, but overall I think this is extremely promising work, but would benefit from more data to demonstrate that the tradeoffs involved are worthwhile at this point in time. It seems likely that the embedded world will head in this direction, but if the tooling, latency, and binary size costs are too high it may be too early, and the best way to figure that out is with case studies and metrics.

**Paper Strengths:**

Strengths listed below, with notes:

### Sensor standardization

I know from experience that writing interfaces to microphones, accelerometers, gyroscopes, magnetometers, and cameras is one of the most time-consuming parts of embedded ML development. Having a single API layer for common sensors that an application can be developed against once and deployed across many devices is a big advance.

### Webassembly portability

Using Webassembly on embedded platforms offers a lot of potential advantages in development, portability, deployment, and debugging, compared to traditional C/C++ workflows.

**Paper Weaknesses:**

Weaknesses listed below, with notes:

### Lack of case studies

As a solution for developers, the test of this approach will be how well it works in practice. Without more examples of different applications (audio recognition, accelerometer gestures, computer vision) implemented using both approaches to contrast and compare the tradeoffs, it's hard to tell if this will be widely accepted.

### Need for metrics

Some of the key characteristics of this solution are unclear to me. For example, the binary footprint (how much space in flash is used by the library code included in an executable) of the solution doesn't seem to be listed, and this is a common constraint for embedded developers, and would determine whether this approach is usable in many cases. It would also be helpful to understand how this would fit into existing toolchains in more detail.

**Poster (If Paper Is Rejected):**

1: Yes, ok for poster sesion to nurture work

**Reviewer Confidence:**

4: The reviewer is confident but not absolutely certain that the evaluation is correct

---

### Decision · Program_Chairs · 2021-02-05

**Decision:**

Accept (Poster)

**Comment:**

Based on the reviewer feedback, your paper has been accepted as a poster.

Please read the reviews carefully and make sure the concerns are addressed in your poster submission.

Accepted posters are given a 5-minute slot for an oral presentation on Friday, March 26, 2021, to pitch the main ideas of your work and to stimulate discussions. Detailed instructions will follow soon. All final posters will earn a stamp of acceptance as such: “Published as a poster at TinyML Research Symposium 2021.”